# Reproducibility Study of "Improving Interpretation Faithfulness For Vision Transformers"

**Meher Changlani**                                          *meher.changlani@student.uva.nl*
*University of Amsterdam*

**Aswin Krishna Mahadevan**                     *aswin.krishna.mahadevan@student.uva.nl*
*University of Amsterdam*

**Benjamin Hucko**                                          *benjamin.hucko@student.uva.nl*
*University of Amsterdam*

**Ioannis Kechagias**                                      *ioannis.kechagias@student.uva.nl*
*University of Amsterdam*

**Reviewed on OpenReview:** *https://openreview.net/forum?id=aOrytDAGUD*

## Abstract

This paper attempts to reproduce the findings of the study "Improving Interpretation Faithfulness For Vision Transformers" Hu et al. (2024). The authors focus on making visual transformers (ViTs) more robust to adversarial attacks, and calling these robust ViTs as faithful ViTs (FViTs). In their paper they propose a universal method to transform ViTs to FViTs called denoised diffusion smoothing (DDS). The reproduction of the authors study suffers from certain challenges, but the main claims still hold. Furthermore, this study extends the original paper by trying different diffusion models for DDS and tries to generalize the increased robustness of FViTs.

## 1 Introduction

Vision Transformers (ViTs) (Ramachandran et al., 2019) have shown remarkable promise in recent years, achieving state-of-the-art performance in computer vision tasks such as image classification (Dosovitskiy et al., 2021), semantic segmentation (Zheng et al., 2021), and object detection (Zhu et al., 2021). Beyond their efficacy, the attention mechanisms in ViTs offer a way to visualize which parts of the input the model focuses on during predictions, which aids with interpretability. However, these very attention mechanisms also make the model particularly susceptible to adversarial attacks (Mahmood et al., 2021; Zheng et al., 2022; Wiegreffe & Pinter, 2019; Hu et al., 2022), as such attacks exploit the model's attention mechanisms and global feature dependencies, leading to incorrect predictions with only minor perturbations to the input.

To address this issue, Hu et al. (2024) introduces a new category of ViTs, called Faithful Vision Transformers (FViTs), that are designed to be resilient to minor input perturbations. Jacovi & Goldberg (2020) defined faithful models as those providing interpretations that reflect the true reasoning process of the model when making a decision. To this end, the authors propose a method, called Denoised Diffusion Smoothing (DDS), for converting a standard ViT into an FViT. In brief, DDS applies randomized smoothing to the input of a standard ViT and then uses a denoised diffusion probabilistic model Ho et al. (2020) to process the perturbed input. By repeating this process multiple times and averaging the attention activations, the authors claim that the ViT can be transformed into an FViT.

The authors Hu et al. (2024) first establish the theoretical effectiveness of DDS and FViTs and then validate it through a series of experiments. These experiments evaluate the performance of various ViTs and their FViT counterparts on classification and segmentation tasks under adversarial attacks and under normal

conditions. In this study, we aim to verify their findings by reproducing their experiments, enhance the robustness of FViTs by improving the DDS algorithm using alternative diffusion models, and extend the experimentation to demonstrate that models produced through DDS are robust, not only to adversarial attacks, but also to various types of perturbations.

## 2 Scope Of Reproducibility

In all their experiments, the authors compared their proposed method, which employs faithful versions of ViTs, with five other baselines 1 that use ViTs without modifications. The comparisons are based on classification accuracy and segmentation metrics (pixel accuracy, mIoU and mAP), as well as the consistency of the generated attention maps under adversarial attack compared to normal conditions. The proposed method transforms a ViT into an FViT using DDS and then applies Visual Transformers Attribution (VTA) (Chefer et al., 2021) as a post-hoc method to generate attention maps. In contrast, the other five baselines are standard ViTs (not converted to FViTs) paired with either raw attention (directly computed during the forward pass) Vaswani et al. (2017) or post-hoc methods such as Rollout (Abnar & Zuidema, 2020), GradCAM (Selvaraju et al., 2017), LRP (Binder et al., 2016), and VTA to generate attention maps.

| Baseline | DDS Used | Method For Generating Attention Maps |
|---|---|---|
| Proposed Method | Yes | VTA |
| Baseline 1 | No | Raw Attention |
| Baseline 2 | No | Rollout |
| Baseline 3 | No | GradCAM |
| Baseline 4 | No | LRP |
| Baseline 5 | No | VTA |

Table 1: Comparison of baselines used for evaluation, specifying the use of DDS and the method for generating attention maps.

This study will verify the following claims of the original paper:

- Claim 1: The DDS algorithm is architecture-agnostic for visual transformers.
- Claim 2: The proposed method outperforms all other considered baselines under adversarial attacks for all metrics across all datasets and models, for both classification and segmentation tasks.
- Claim 3: The heatmap visualizations generated by the proposed method exhibit greater consistency compared to all other baselines under adversarial attacks, across all datasets and models, for both classification and segmentation tasks.
- Claim 4: The proposed method outperforms all other considered baselines for both negative and positive perturbation tests.

and the following claims that extend the original paper:

- Claim 5: The DDS algorithm performs comparably when combined with diffusion models working on the latent space.
- Claim 6: Faithful models exhibit comparable or inferior performance to their non-faithful counterparts when classifying corrupted images that do not result from adversarial attacks.

## 3 Methodology

To replicate the study, we utilized the public repository provided by the authors, which includes code for the baselines, post-hoc methods, models, and helper functions for loading datasets. Our adapted code can be found on GitHub.

### 3.1 Model descriptions

#### 3.1.1 ViT Models

' **Vanilla ViT.** The original Vision Transformer model (Dosovitskiy et al., 2021) treats images as a series of smaller patches and uses the transformer architecture to process them for vision tasks. The attention values are used to generate attention maps using relevancy propagation (Chefer et al., 2021).

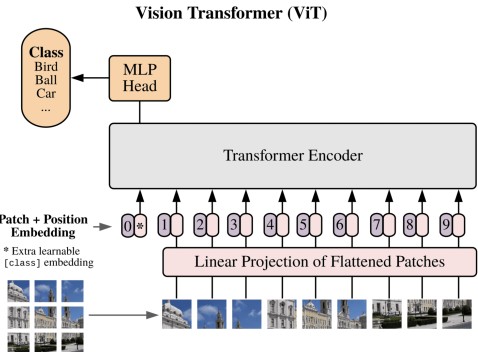

Figure 1: Visualization of the Vanilla ViT architecture taken from Dosovitskiy et al. (2021).

**DeiT.** A more efficient version of ViT (Touvron et al., 2021), which performs well with less training data by using a method called knowledge distillation, where a smaller model learns from a larger pre-trained model.

**Swin ViT.** A variant of ViT (Liu et al., 2021) that uses a hierarchical structure and "shifted windows" to process images more effectively, capturing both detailed and broader image features.

#### 3.1.2 Diffusion Models For DDS

Diffusion models are able to generate images by repeated sampling of $p_\theta(x_{t-1} \mid x_t)$ (see figure 2) from noise $x_T$ to generated image $x_0$ (Ho et al., 2020). The probability distribution is a Gaussian Distribution parametrized by

$$N\left(\boldsymbol{\mu}_\theta(\boldsymbol{x}_t, t), \boldsymbol{\Sigma}_\theta(x_t, t)\right),$$

where $\boldsymbol{\mu}_\theta(\boldsymbol{x}_t, t), \boldsymbol{\Sigma}_\theta(\boldsymbol{x}_t, t)$ are approximated by a U-NET network (Ronneberger et al., 2015). The full algorithm behind denoising is in the Appendix. To decrease computational requirements, Ho et al. (2020) provide a formula to approximate $\boldsymbol{x}_0$ analytically from $\boldsymbol{x}_t$ and $\boldsymbol{x}_{t-1}$.

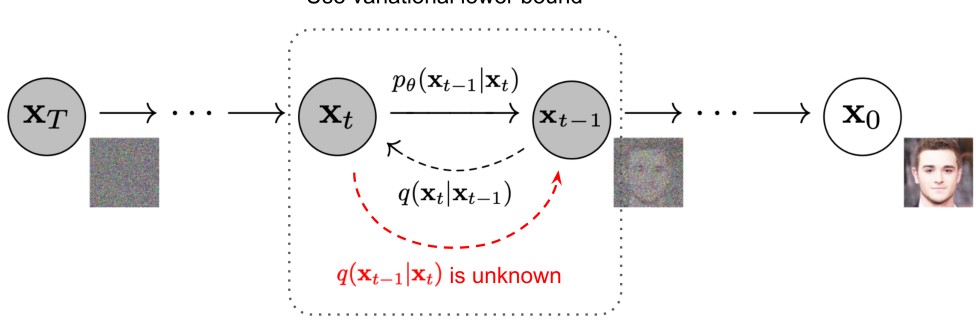

Figure 2: Markov chain schematic of diffusion model taken from Ho et al. (2020). $x_t$ is the noisy sample generated with the randomized smoothing, $x_0$ is the denoised sample. $q(x_t \mid x_{t-1})$ is the noisy sample distribution and $p_\theta(x_{t-1} \mid x_t)$ is the parametrized approximation of the denoised sample distribution.

**Guided-Diffusion**. Guided diffusion models (Ho et al., 2020) integrate diffusion models with guidance mechanisms, such as labels or gradients, to steer the generation process and enhance output control. For the ImageNet dataset, we utilized the pre-trained diffusion model provided by guided-diffusion as the denoiser.

**Latent-Diffusion (Extension)**. Latent diffusion models (LDMs) (Rombach et al., 2022) combine diffusion models with an autoencoder. The model encodes the image, and then performs the diffusion steps on the latent space. After diffusion, the latent space is decoded back into the image space. LDMs inherently require less memory and compute as compared to pixel-space models, while also maintaining high quality and flexibility of applications.

**Stable-Diffusion (Extension)** Stable diffusion models (SD) are a subset of LDMs and guided diffusion models know for their accessibility and state-of-the-art performance. They perform diffusion in the latent space using text prompts to guide the diffusion process.

### 3.1.3 Classification Datasets

**ILSVRC-2012 ImageNet.** A large-scale dataset widely used for image classification and object recognition tasks. It consists of 1.28 million training images, 50,000 validation images, and 100,000 test images, spanning 1,000 object categories. Each image is annotated with a single label corresponding to one of these categories.

**ImageNet-C (Extension).** Hendrycks & Dietterich (2019). This dataset consists of 15 diverse corruption types applied to the validation images of the ImageNet dataset. The corruptions are mainly drawn from four categories - noise, blur, weather and digital. Each of these categories has further divisions into subcategories. We used 10,000 images each from the *defocus blur* subcategory within blur and the *elastic transformation* subcategory within digital. Defocus blur occurs when an image is out of focus and elastic transformation stretches or contracts images.

### 3.1.4 Segmentation Datasets

**ImageNet-Segmentation Subset.** A subset of the ImageNet dataset (Guillaumin et al., 2014) that provides pixel-level object-background segmentations for 500,000 images spanning 577 object categories.

**COCO.** Also known as "Common Objects in Context" dataset (Lin et al., 2015) is a large-scale image recognition, segmentation, and captioning dataset. It contains over 330,000 images, with more than 200,000 labelled, spanning 80 object categories. The dataset includes annotations for object detection, segmentation, keypoint detection, and image captions.

**Cityscapes.** The Cityscapes dataset (Cordts et al., 2016) focuses on semantic understanding of urban street scenes and includes 5,000 annotated images with fine annotations and 20,000 annotated images with coarse annotations, spanning 30 classes.

## 3.2 Implementation

### 3.2.1 Denoised Diffusion Smoothing (DDS)

The DDS algorithm comprises two key steps: smoothing and denoising. Initially, noise is introduced and scaled in proportion to the number of diffusion steps. Subsequently, the image undergoes denoising using a diffusion model for an optimal number of iterations, denoted as $t^*$ By incorporating DDS before the forward pass, a standard ViT is transformed into a FViT.

---

**Algorithm 1** Denoised Diffusion Smoothing (DDS)

---

**Input:** Image $\boldsymbol{x}$, noise level $\delta$, diffusion steps $N$, schedule bounds $\beta_1$, $\beta_2$.
**Flags:** Denoising (default: True), Smoothing (default: True).
**Output:** A denoised and smoothed image $\hat{\boldsymbol{x}}$.
$t^* \leftarrow \texttt{get\_optimal\_number\_of\_steps}(\delta, \beta_1, \beta_2, N)$      ▷ Optimal step selection.
$\beta_{t^*} \leftarrow \beta_1 + \dfrac{t-1}{N-1}(\beta_2 - \beta_1)$      ▷ Noise Schedule.
$\boldsymbol{x}_{t^*} \leftarrow \sqrt{1 - \beta_{t^*}}(\boldsymbol{x} + \epsilon)$ where $\epsilon \sim \mathcal{N}(0, \delta^2)$      ▷ Noise injection.
$\hat{\boldsymbol{x}} \leftarrow \texttt{diffusion\_denoise}(\boldsymbol{x}_{t^*}, t^*)$      ▷ Denoising via diffusion model.
$\hat{\boldsymbol{x}} \leftarrow \hat{\boldsymbol{x}} + \epsilon'$ where $\epsilon' \sim \mathcal{N}(0, \delta^2)$
**return** $\hat{\boldsymbol{x}}$

---

Note that the final smoothing step (post-denoising) was present in the original implementation but not explicitly mentioned in the paper. Additionally, we assume that Hu et al. (2024) used this analytical approximation, as they report computation time independent of $T$. This significantly improves runtime without substantial drop in performance.

### 3.2.2 FViT

The FViT offers two key improvements over the standard ViT: enhanced interpretability of attention activation maps and increased robustness under adversarial attacks. These improvements are achieved by evaluating the ViT model on multiple DDS-processed input samples and subsequently aggregating the corresponding predictions. This ensures more stable and reliable decision-making and improves model transparency.

---

**Algorithm 2** FViT

---

**Input:** image $\boldsymbol{x}$, num trials $m$, ViT model $y$, dds hyperparameters $\boldsymbol{\theta}$.
**Output:** FViT output as probability distribution $\boldsymbol{c}_{aggregate}$, FViT attention map $\boldsymbol{w}_{aggregate}$.
**for** $m$ **do**
     $\hat{\boldsymbol{x}} \leftarrow \text{dds}(\boldsymbol{x}, \boldsymbol{\theta})$
     $\boldsymbol{c}_i, \boldsymbol{w}_i \leftarrow y(\hat{\boldsymbol{x}})$
**end for**
$\boldsymbol{w}_{aggregate} \leftarrow \frac{1}{m} \sum_{i=1}^{m} \boldsymbol{w}_i$
$\boldsymbol{c}_{aggregate} \leftarrow \text{aggregate}(\boldsymbol{C})$
**return** $\boldsymbol{c}_{aggregate}, \boldsymbol{w}_{aggregate}$

---

Hu et al. (2024) performed the aggregate for the model output $c$ by taking the top prediction for each of the $m$ timesteps and transforming these $m$ prediction into a probability distribution,

$$\boldsymbol{c}_{aggregate}^{(j)} \leftarrow \frac{\sum_i^m \text{argmax } \boldsymbol{c}_i = j}{m},$$

where $\boldsymbol{c}_{aggregate}^{(j)}$ corresponds to the index $j$ of aggregated probability distribution. We suggest a new aggregation method, where we keep the output $c_i$ as the probability distribution over the classes and aggregate them into $c$ by taking the average,

$$\boldsymbol{c}_{aggregate} \leftarrow \frac{1}{m} \sum_i^m \boldsymbol{c}_i.$$

We hypothesize that our method will have more descriptive top-k accuracy for $k \neq 1$, because the aggregation method suggested by Hu et al. (2024) should struggle with saturated probability distribution. This is because at most $m$ classes will have $p > 0$, and on average even less classes will have $p > 0$. Therefore the top-k accuracies will not be relevant if $k > \text{count}_i(p_i > 0)$. We show experimentally in Appendix table 6 that our mean aggregation drastically improves Top-5 accuracy.

### 3.3 Experiments

**Experiment 1 (Claims 1, 2):** Evaluate all the ViTs on the classification task using the ILSVRC-2012 ImageNet dataset and compare their performance. For each baseline, compute the following metric: classification accuracy.

**Experiment 2 (Claims 1, 2):** Evaluate the performance of all ViTs across all baselines on the segmentation task using the ImageNet segmentation dataset. To assess interpretability, compare each model's explanation maps against the actual object regions, using segmentation labels as the ground truth, following the approach outlined by Chefer et al. (2021). Interpretability is measured using pixel accuracy, mean intersection over union (mIoU) (Varghese et al., 2020), and mean average precision (mAP) (Henderson & Ferrari, 2017). Specifically, pixel accuracy is calculated by thresholding the visualization using the mean value, while mAP utilizes soft-segmentation to generate a threshold-independent score.

**Experiment 3 (Claims 3, 4):** Conduct positive and negative perturbation tests for all ViTs across all baselines under default attack positions while varying the attack radius from 0 to 32/255. In detail, first generate visualizations for each attribution method and save them to corresponding hdf5 files, which are used to calculate the highly relevant and irrelevant pixels for each image. In positive perturbation, pixels are masked from the highest relevance to the lowest, while in the negative version, from lowest to highest. A lower score in positive perturbations indicate that the model relies on important pixels and a high score in negative perturbations indicates that it does not rely on unimportant pixels. For each baseline attribution method, compare the AUC (Area-under-curve) of the accuracy vs pixel perturbation percentage (ranging from 0-90%) plot for every attack radius. This experiment is used to highlight whether DDS makes the models consistently rely more on important pixels and less on irrelevant pixels. This acts as a quantitative analysis proving that FViTs have consistent and accurate reasons backing their decisions.

**Experiment 4 (Claims 3, 4):** Visualize the model heatmaps for the original and perturbed images. Observe how these attention heatmap changes when focusing on one class present in the image compared to the other class present in the image. Determine that the model is not class agnostic, i.e. has a heatmap independent of the class label. These findings act as qualitative analysis proving that FViTs have consistent and accurate reasons backing their decisions.

### 3.4 Extension

**Extension 1 (Claim 5)**

We extend the original work by performing DDS with diffusion models working in the latent space to explore the versatility of the DDS algorithm. These models achieve comparable, and in some cases better, performance in image generation benchmarks with lower computational cost (Rombach et al., 2022). However, the effect of the PGD attack on the original image has unpredictable consequences when the encoder transforms the image into the latent space. Therefore, the claims about DDS need to be reevaluated for diffusion models applied to the latent space. For this purpose, we test DDS with latent diffusion models (LDMs), stable diffusion (SD) Rombach et al. (2022), and larger models like XLSD (Podell et al., 2023) and Kandinsky (Razzhigaev et al., 2023). We use the class-conditioned LDM pre-trained on ImageNet and a zero vector as the label embedding. We choose the text-conditioned SD pre-trained on Laion aesthetics dataset and used the prompt "clean and detailed imagenet image".

**Extension 2 (Claim 6)**

Lastly, we believe that PGD attacks are a limited metric when it comes to proving that DDS improves the robustness of ViTs. The PGD uses the gradients of the model with respect to the input image to move it in the direction of the decision boundary of the ViT. In the case of DDS, the input is then fed into the diffusion model. If the diffusion model has a region of inputs for which it generates an image of a certain class, the PGD attacked image is as likely to still be in that region as image with random noise. Therefore, this method is biased against the baseline. This leads us to performing classification on other benchmarks with independent deviations. For this purpose we chose the ImageNet-C dataset, which consists of naturally

| Model | With DDS | Without DDS |
|-------|----------|-------------|
| ViT-B | 72.44 | 8.33 |
| DeiT-B | 72.25 | 4.77 |
| Swin-T | 72.50 | 3.17 |

Table 2: Top-1 classification accuracy on the ILSVRC-2012 validation dataset with default attack.

corrupted images. The corruption in ImageNet-C are not biased towards DDS, because the corruption also affects the generation of diffusion models.

# 4 Results

## 4.1 Reproducibility

First, we evaluated the proposed method on a classification task under a PGD attack. The results in table 2 show that when the models do not use DDS they achieve very low accuracy (below 10%), whereas when the models use DDS they achieve accuracy exceeding 72%.

Next, we evaluated the DDS method and baseline models on a segmentation task under adversarial attacks. Table 3 shows the DDS method outperformed all baselines across all metrics and for all models. This demonstrates that the DDS algorithm successfully transforms ViTs into FViTs under the definition of faithfulness provided by Hu et al. (2024). Since this trend holds for two out of the three Vision Transformer architectures explored by the authors (ViT and DeiT) we can conclude that DDS is architecture-agnostic for attention-based models. Based on these findings, we confirm that Claims 1 and 2 hold true.

| Model | Method | Pix. Acc. | mIoU | mAP |
|-------|--------|-----------|------|-----|
| ViT-B | Raw Attention | 0.63 | 0.42 | 0.77 |
| | Rollout | 0.73 | 0.55 | 0.84 |
| | GradCAM | 0.68 | 0.42 | 0.68 |
| | LRP | 0.51 | 0.33 | 0.55 |
| | VTA | 0.73 | 0.52 | 0.82 |
| | VTA (with DDS) | **0.76** | **0.65** | **0.85** |
| DeiT-B | Raw Attention | 0.68 | 0.47 | 0.80 |
| | Rollout | 0.59 | 0.39 | 0.73 |
| | GradCAM | 0.66 | 0.41 | 0.69 |
| | LRP | 0.51 | 0.32 | 0.55 |
| | VTA | **0.79** | 0.61 | 0.85 |
| | VTA (with DDS) | **0.79** | **0.62** | **0.86** |

Table 3: Performance comparison of different methods on the Imagenet Segmentation dataset under the default attack.

Furthermore, we conducted the positive and negative perturbation tests, as shown in Figure 4.1. Our findings indicate that their results are only partially reproducible.

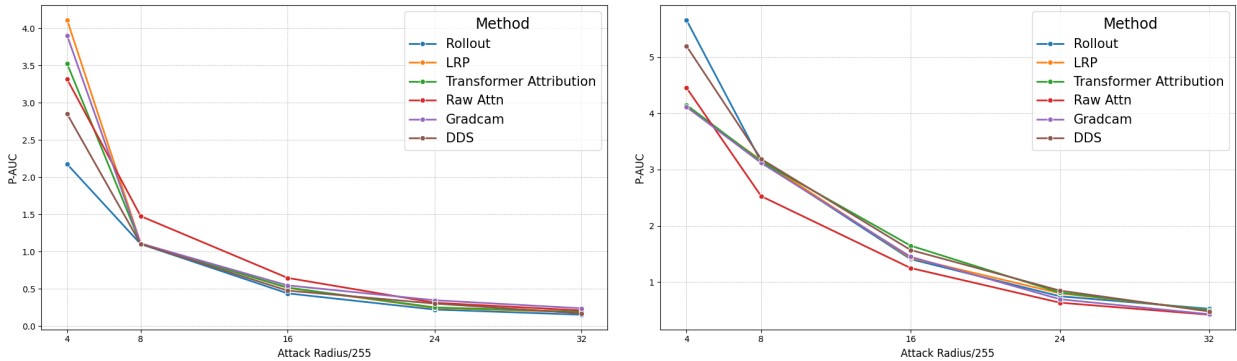

Figure 3: Positive Perturbation.       Figure 4: Negative Perturbation.

The positive perturbation test shows that DDS achieves the lowest P-AUC for most attack radii. This supports the claim that the model focuses on important pixels. However, our results reveal that P-AUC decreases as the attack radius increases, which seems more intuitve and contradicts the findings in the original paper, where P-AUC increases with attack radius. A higher attack radius implies a stronger attack, causing a higher deviation. A stronger attack can never have a better classification performance for any method. Based on our implementation, we find the outcome of the original paper highly counterintuitive and find our results more realistic.

On the other hand, our negative perturbation test fully aligns with the original paper. The DDS consistently outperforms other methods across most attack radii. This confirms that DDS relies the least on unimportant pixels. This successfully validates Claims 3 and 4, demonstrating that DDS consistently generates accurate importance maps and prioritizes important pixels in its decisions.

We also conducted a qualitative analysis of attention maps on several samples containing at least two objects, where some objects were more dominant than others. We ran two experiments: first, focusing on the dominant objects, and then on the less dominant ones.

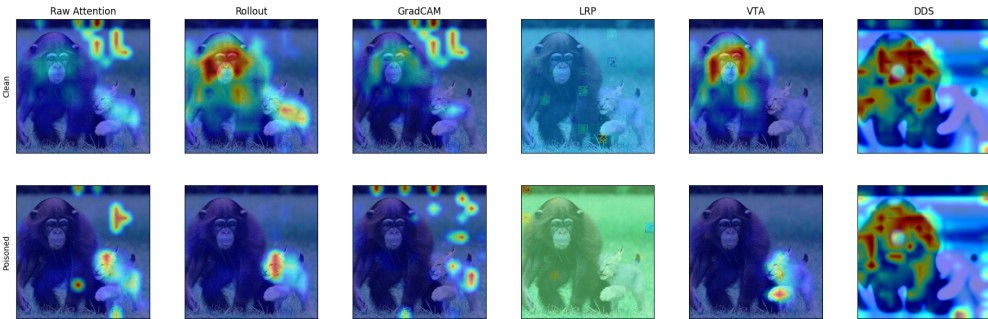

Figure 5: Baseline methods used to interpret the probability behind class the monkey.

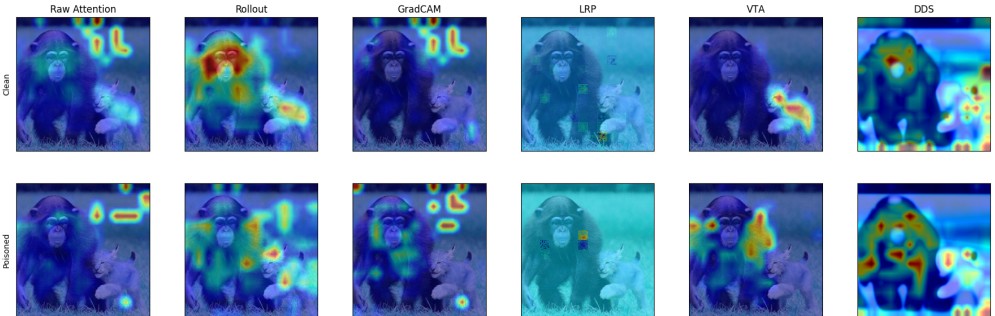

Figure 6: Baseline methods used to interpret the probability behind the class lynx.

In the first case, the DDS algorithm performed significantly better than the baselines when the image was attacked (Figure 5). However, when the focus shifted to the less dominant objects, DDS performed significantly worse than the VTA baseline. For example, in Figure 6, where the focus is on the lynx instead of the monkey, only VTA produced a reasonable heatmap for the non-attacked image, while none of the other methods generated a consistent and logical heatmap. This suggests a potential limitation of DDS in explaining classifications for non-dominant classes within an image.

## 4.2 Extensions

### 4.2.1 Using Different Diffusion Models With DDS (Extension)

When running DDS with different diffusions, we observed that the performance of the DDS algorithm degraded, as shown in Table 4. Despite the reduced performance, LDM and SD still significantly outperform the baseline of not using DDS. Therefore, we conclude that diffusion models working on the latent space still help the model achieve robustness against adversarial attacks similarly to when using diffusion model working in the image space. This shows us that LDMs can be used with DDS and confirms Claim 5. On top of that, both LDM and SD had a shorter computation time per sample. We view the choice between using the original diffusion model, SD and LDM for DDS as a trade-off between performance and computation time. Finally, we do not see the benefit of using larger diffusion models, such as SDXL and Kandinsky, because these have too long computation time.

| Model | Accuracy [%] | Per-sample computation time [ms] |
|---|---|---|
| Hu et al. (2024) | 72.44 | 359 |
| Stable Diffusion XL | 70.56 | 819 |
| Stable Diffusion 1.4 | 65.66 | 313 |
| Latent Diffusion | 56.09 | 283 |
| Kandinsky 2.2 | 52.38 | 7401 |
| No DDS | 7.3 | 60 |

Table 4: Top-1 Accuracy on the ILSVRC-2012 validation dataset with attack.

### 4.2.2 Evaluating on ImageNet-C (Extension)

For our second extension, we investigate the performance of the ViT on a new benchmark, the ImageNet-C dataset. The results displayed in 5 show that using DDS reduces accuracy on the ImageNet-C dataset, indicating that DDS does not enhance robustness for blurred images and those subjected to elastic transformations. This proves Claim 6 that DDS, while architecture agnostic, heavily depends on the attack parameters being well defined and known.

| Corruption Category | With DDS | Without DDS |
|---|---|---|
| Defocus Blur | 62.75 | 74.22 |
| Elastic Transformation | 3.76 | 3.90 |

Table 5: Top-1 Accuracy on the ImageNet-C dataset.

## 5 Discussion

### 5.1 Reproduction challenges

#### 5.1.1 Implementation

We observe that there were discrepancies in the implementation that hindered our reproduction of the results. Despite these challenges, we were able to address several shortcomings wherever possible.

For the ImageNet dataset, the authors used diffusion model weights provided by Ho et al. (2020), whereas for the COCO and Cityscapes datasets, the authors trained their own diffusion models but did not make these models publicly available. As a result, we were unable to reproduce their results for the COCO and Cityscapes datasets. Even the provided environment was missing several packages or had compatibility issues between the versions of dependencies, making the initial setup for the reproducibility a tedious task.

Additionally, the scripts they inherited from Chefer et al. (2021) were not available in the author's repository or even mentioned in the README, so we had to manually copy and integrate them into our source code. We also had to implement the ImageNet classification script, as it was missing from the original repository. Additionally, we improved both the DDS and baseline models to run batch-wise, reducing computation time by approximately 66%.

Lastly, neither the API nor the architecture of the Swin Transformer that was compatible with their implementation was provided, preventing us from replicating the segmentation experiments with Swin. Crucially, the DDS implementation in the original repository provided by the authors performed an additional randomized Gaussian smoothing step after the diffusion denoising step, which did not align with the approach described in the original paper.

#### 5.1.2 Methodology

The authors get different classification results for different interpretation baselines. However, the algorithm in the paper suggests that the classification is independent of interpretation method. Therefore, in our reproduction the classification has the same accuracy for all interpretability baselines.

Hu et al. (2024) do not clearly describe how they calculated the P-AUC and how they performed the perturbation attack. For the P-AUC they write that it is the area under curve of accuracy based on percentage of removed pixels. The authors specify that the removed pixels range from 10% to 90%, but do not give how many samples they gathered. They also fail to specify in what context is attack radius defined for this experiment, as perturbation tests in general are done without any adversarial attack.

#### 5.1.3 Communication with original authors

We tried contacting the authors using two different means. Firstly, we created a GitHub issue which contained our questions regarding their implementation, with no response from their side. Secondly, we emailed our questions to the authors, who responded that the code they shared on GitHub is an old version and outdated. Unfortunately, they have not provided the up-to-date code or responded to our technical questions yet despite multiple follow-ups.

## 5.2 Limitations

Our reproducibility study was constrained by limited computational resources. As a result, we were unable to perform as many experiments as we would have liked, which weakens the robustness of our findings.

### 5.2.1 Reproduction study

We did not evaluate whether adding additional noise after denoising enhances the performance of the DDS algorithm due to lack of clarity. Thus, we evaluated the implementation of the DDS algorithm in the official code base but not the algorithm as described in the paper. This introduces a potential discrepancy in our reproduction, which must be addressed for a fair evaluation of DDS.

Furthermore, we did not compare the two aggregations for model predictions under DDS and just used $c_{aggregate} \leftarrow \frac{1}{m} \sum_i^m c_i$. We believe that the theoretical and intuitive foundation behind our aggregation is enough to justify using it.

### 5.2.2 Our Extension

Owing to resource constraints, we were unable to train an LDM that performs unconditional denoising. Consequently, the LDM we used was not trained for the exact task we applied it to, which limits the reported performance of LDM. However, this does not affect the conclusion in this extension.

Additionally, we only tested DDS on a subset of the ImageNet-C dataset. Due to the significant computational demands of the reproduction study, we lacked the resources to evaluate DDS across the entire ImageNet-C dataset.

### 5.3 Future research

As part of further research, we can experiment with more state-of-the-art interpretability metrics like the Salience-guided Faithfulness Coefficient (SaCo) (Wu et al., 2024), which claims that it significantly enhances attention-based explanation. Combining SaCo with DDS should improve the state-of-the-art performance.

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

# A   Appendix

## A.1   Algorithms

---

**Algorithm 3** Diffusion Denoising

**Input:** Image $x_T$, timestep $T$.
**Output:** A denoised image $x_0$.
**for all** $t \in \{1, \ldots, \text{T}\}$ **do**
    $\boldsymbol{\mu} \leftarrow \boldsymbol{\mu}_\theta(\boldsymbol{x}_t, t)$
    $\boldsymbol{\Sigma} \leftarrow \boldsymbol{\Sigma}_\theta(\boldsymbol{x}_t, t)$
    **if** $(t-1) \neq 0$ **then**
        $\boldsymbol{x}_{t-1} \sim N(\boldsymbol{\mu}, \boldsymbol{\Sigma})$
    **else**
        $\boldsymbol{x}_{t-1} \leftarrow \boldsymbol{\mu}$
    **end if**
**end for**
**return** $\boldsymbol{x}_0$

---

**Algorithm 4** Optimal Step Selection (`get_optimal_number_of_steps`)

**Input:** Noise level $\delta$, schedule bounds $\beta_1$, $\beta_2$, total steps $N$.
**Output:** Optimal number of steps $t^*$.
$t^* \leftarrow \left\lfloor 1 + \frac{N-1}{\beta_2 - \beta_1} \left( 1 - \frac{1}{1+\delta^2} - \beta_1 \right) \right\rceil$
**return** `clip`$(t^*, 0, N)$                                 $\triangleright$ Ensure $t^* \in [1, N]$.

---

### A.2  Environmental Impacts

We estimated the carbon footprint of our research using

$$CO_2e = CI \cdot PUE \cdot P \cdot t,$$

where $CO_2e$ are the Carbon emissions; $CI$ corresponds to the Carbon intensity measured in grams of carbon dioxide-equivalents emitted per kilowatt-hour of electricity generated; $PUE$ is the Power Usage Effectiveness of the used cluster; $P$ is the power usage of the CPU, GPU and memory; and $t$ is the computation time.

For all our experiments we used using a single GPU, an Nvidia A100 80GB. We ran 212 experiments for a total of 316 hours or roughly 2 weeks continuous, and spent 81.58 kW equaling a total of 110.72kWh of energy spent. We use the $CI$ value of $370 gCO2eq/kWh$ for the Netherlands in 2024 obtained from Nowtricity and $PUE$ value of 1.3 according to State of the Dutch Data Centers Report 2024. This brings the total carbon emissions to $53.26 kg$.

However this value would have been much higher using the implementation provided by the authors due to the inefficiencies of the official code base. We extrapolated the runtime and energy spent and estimated that the CO2 emissions in case we had ran all the experiments using the original code would be $\approx 97.82 kg$. This translates to $\approx 45.6\%$ reduction in CO2 emissions stemming from our code optimization.

### A.3  Reduction Methodology Comparison

| Aggregation Method | Top-1 | Top-5 |
|---|---|---|
| Mean (Ours) | 56.09 | 79.81 |
| Max | 55.14 | 55.26 |

Table 6: Performance comparison of different methods on classification accuracy on the ILSVRC-2012 validation dataset using Latent diffusion with default attack.

