# OpenReview forum: "Reproducibility Study of "Improving Interpretation Faithfulness For Vision Transformers""
_TMLR — Accepted by TMLR_

### Review · Reviewer_Q43y · 2025-03-18

**Summary Of Contributions:**

The paper attempts to reproduce the findings of the study "Improving Interpretation Faith-fulness For Vision Transformers" Hu et al. (2024). The authors focus on making visual transformers (ViTs) more robust to adversarial attacks, and calling these robust ViTs faithful ViTs (FViTs). In their paper they propose a universal method to transform ViTs to FViTs called denoised diffusion smoothing (DDS). The reproduction of the authors study suffers from certain challenges, but the main claims still hold. Furthermore, this study extends the original paper by trying different diffusion models for DDS and tries to generalize the increased robustness of FViTs.

**Audience:**

Yes

**Claims And Evidence:**

Yes

**Requested Changes:**

1. getOptimalNumberOfIterations not clear; did not present details of how to compute

2. latent space PGD experiment not clear; whether the perturbation launch on latent embedding or on the pixel is not clear;

3. The appendix presentation needs to be polished; some format error

4. The figure1 is from vit, please add a reference in the caption.

**Strengths And Weaknesses:**

weakness:

1. getOptimalNumberOfIterations not clear; did not present details of how to compute

2. latent space PGD experiment not clear; whether the perturbation launch on latent embedding or on the pixel is not clear;

3. The appendix presentation needs to be polished; some format error

4. The figure1 is from vit, please add a reference in the caption.

strength

1. On the model output aggregation of FViT, a new average aggregation method is proposed in order to obtain more descriptive top-k accuracy results. This optimization exploration of method details helps to further improve the model performance and the rationality of result interpretation. However, the reasoning in the text should be more clear;

2. This work extends the robustness study of FViT on the latent space diffusion model and natural perturbation. Although the results indicate limitations in its robustness in FViT, this exploration helps to comprehensively understand the applicability and potential issues of DDS, pointing the way for subsequent research.

3. Based on the DDS method and FViT proposed by Hu and others, through systematic reproduction and extensional research, its effectiveness, versatility, and limitations have been verified with 6 proposed claims, providing clear directions and references for research in this field.

---

> ### Author Response · Authors · 2025-05-14
> **Thanks and response to concerns**
>
> > getOptimalNumberOfIterations not clear; did not present details of how to compute
>
> We added the calculation for the optimal number of iterations as algorithm 4 in the appendix.
>
> > Latent space PGD experiment not clear; whether the perturbation launch on latent embedding or on the pixel is not clear;
>
> We rewrote the section to make it clear that the perturbation is done on the pixel space.
>
> > The appendix presentation needs to be polished; some format error
>
> We improved the structure and formatting of the appendix.
>
> > The figure1 is from vit, please add a reference in the caption.
>
> We added the reference in the caption of figure 1 to the ViT paper.

---

### Review · Reviewer_xjQS · 2025-03-26

**Summary Of Contributions:**

This paper presents a reproducibility study of Hu et al. (2024), which proposed a method called Denoised Diffusion Smoothing (DDS) to enhance the interpretability and robustness of Vision Transformers (ViTs) under adversarial attacks. The original work transforms standard ViTs into Faithful ViTs (FViTs), and the reproduced study verifies and extends several of their key claims. The key contributions of the reproducibility study include:
1.Verification of Original Claims.
2.Methodological Enhancements:Evaluation of DDS using alternative diffusion models, such as Stable Diffusion and Latent Diffusion Models (LDMs). Testing on ImageNet-C, to assess generalization of robustness beyond adversarial settings.

**Audience:**

Yes

**Claims And Evidence:**

Yes

**Requested Changes:**

Incorporate Stronger Diffusion Models in DDS Experiments: recent advancements in diffusion modeling introduce architectures that may significantly improve DDS performance, maybe the author can test those methods in their experiments. For example, DIT, SD3, SANA.

**Strengths And Weaknesses:**

Strengths

Thorough Reproduction: The authors meticulously reproduce the core experiments of Hu et al. (2024) and provide credible validation for most claims.

Technical Rigor: Extensions and deviations are backed by well-reasoned justifications (e.g., in aggregation methods and metric definitions).

Novel Insights: The observation that DDS may overfit to adversarial attack styles while underperforming on non-adversarial corruptions is important and novel.

Weakness

Some figures in the paper (e.g., Figure 1) appear to be identical to those from the original ViT paper,  is this comply with TMLR’s publication policy?

---

> ### Author Response · Authors · 2025-05-14
> **Thanks and response to concerns**
>
> > Incorporate Stronger Diffusion Models in DDS Experiments: recent advancements in diffusion modeling introduce architectures that may significantly improve DDS performance, maybe the author can test those methods in their experiments. For example, DIT, SD3, SANA.
>
> We performed DDS experiments with Kandinsky 2.2 and Stable Diffusion XL, both of which are significantly larger models known for their high-quality image generation capabilities. Crucially, these models offer native support for image-to-image diffusion, which allowed us to integrate them into our pipeline. In contrast, models like Stable Diffusion 3 (SD3) currently require external adapters to enable image-to-image diffusion, making them less straightforward to incorporate into our setup. We introduced these in the Methods section and have included the corresponding experimental results into the Results sections.

---

### Review · Reviewer_wFDK · 2025-05-05

**Summary Of Contributions:**

This paper aims to reproduce the main findings of "Improving Interpretation Faithfulness for Vision Transformers" by Hu et al. (2024) (hereafter referred to as the original paper), and further extend and improve upon their methods. The core content of the work includes an analysis of the robustness and interpretability of Vision Transformer (ViT) models under adversarial attacks. The main contributions are as follows:

1. Compared with the original paper, this study not only reproduces the experiments on ViT but also includes DeiT and Swin ViT. The results demonstrate that the conclusions of the original work are generally applicable across different types of ViTs.

2. In addition to the original methods, this paper tests other approaches such as Stable Diffusion, showing that while different diffusion-based models yield consistent conclusions, their performance is not as strong as the Diffusion method used in the original paper.

3. It is observed that the method proposed in the original paper performs poorly on corrupted images such as those in ImageNet-C.

**Audience:**

Yes

**Claims And Evidence:**

Yes

**Requested Changes:**

1. Please see the weakness for further improvement

2. Could you please further explain why the proposed new aggregation method is better than the original one. For example, what is the meaning of saturated probability distribution and is there any evidence to show that the original aggregation does struggle with saturated probability distribution. Moreover, why the new aggregation method has more descriptive top-k accuracy?

**Strengths And Weaknesses:**

## Strengths:
1. The authors clearly explain the background knowledge and provide a relatively clear description of the reproduction methodology and process.
2. The paper is well-written and easy to follow.
3. A new average aggregation method is proposed in order to obtain more descriptive top-k accuracy results.

## Weaknesses

The main goal of reproducibility studies is to provide reliable methods, code, and results for works that are difficult to reproduce, and to derive additional valuable insights based on the original study. However:

1. **The original paper "Improving Interpretation Faithfulness for Vision Transformers" is already open-sourced**. Although some scripts are missing, the methods described in the original paper are clear, making reproduction neither particularly difficult nor labor-intensive. As such, the contribution of this paper as a reproducibility study is limited. Furthermore, the algorithms reproduced here—including DeiT, Swin, and Stable Diffusion—are all accompanied by officially released models and code, reducing the complexity and effort required for adaptation.

2. While this paper does present some additional findings beyond those in the original work, **these findings are rather trivial and do not offer any surprising insights for researchers in the field**. For example, it is expected that the FViT perturbation algorithm would perform similarly across ViT, DeiT, and Swin models.

3. Regarding the experimental results, **the authors did not provide more insightful hypotheses or validations compared to the original paper**. For instance, there is no deeper exploration of why FViT performs poorly on ImageNet-C or why the Diffusion model used in the original work outperforms Stable Diffusion.

4. On page 6, section 3.4 (Extensions), sub-section 1, second line, there is a redundant phrase "computational cost," which appears to be a typographical error.

In conclusion, while this paper reproduces Improving Interpretation Faithfulness for Vision Transformers and offers some extended findings, **these conclusions are relatively trivial, and the reproduction effort is not particularly challenging**. Therefore, the work reads more like an extension or supplementary experiment section of the original paper rather than a standalone, publishable research contribution.

---

> ### Author Response · Authors · 2025-05-14
> **Thanks and response to concerns**
>
> > The original paper "Improving Interpretation Faithfulness for Vision Transformers" is already open-sourced. Although some scripts are missing, the methods described in the original paper are clear, making reproduction neither particularly difficult nor labor-intensive. As such, the contribution of this paper as a reproducibility study is limited. Furthermore, the algorithms reproduced here—including DeiT, Swin, and Stable Diffusion—are all accompanied by officially released models and code, reducing the complexity and effort required for adaptation.
>
> We would like to clarify the challenges we encountered while attempting to reproduce the original work. The publicly available repository associated with the original paper was outdated and not representative of the version used in the published results — a fact we confirmed directly with the original authors. While some scripts were sourced from the Transformer Explainability repository, these were neither documented nor referenced in a way that made them easily findable. Notably, the flag enabling the analytical solution to the diffusion process was absent from the repository leading us to reverse-engineer the implementation to enable the flag in order to match the reported runtime.
>
> Additionally, the environment was incompatible, with several required packages missing. The codebase lacked a dedicated implementation for testing DDS; only a Jupyter notebook for qualitative analysis was provided. We also observed substantial redundancy, with identical code fragments repeated across the repository, all of which required extensive manual refactoring.
> In summary, we believe that our reproducibility study offers significant value. Beyond analyzing the method, we provide a fully functional, refactored and documented implementation of DDS, which we hope will serve as a reliable foundation for future research.
>
> > While this paper does present some additional findings beyond those in the original work, these findings are rather trivial and do not offer any surprising insights for researchers in the field. For example, it is expected that the FViT perturbation algorithm would perform similarly across ViT, DeiT, and Swin models.
>
> While the original authors demonstrated that DDS performs consistently across different vision transformer architectures such as the ViT, DeiT and Swin, our work extends this evaluation by showing that DDS also exhibits similar performance across different diffusion models. This is meaningful because we explore the removal of Gaussian noise-based attacks applied in the pixel space using diffusion models that operate in the latent space. Since Gaussian noise in the pixel space does not translate to Gaussian noise in the latent space, this assumption must be empirically validated rather than taken for granted.
> Furthermore, our qualitative analysis offers valuable insights. We highlight that while DDS enhances faithful regions corresponding to dominant classes, it can introduce increased noise in the faithful zones for less dominant classes – a nuance not discussed previously.
> Lastly, we argue that evaluating DDS under a white-box attack setting is flawed, as DDS involves a different model from the one used in the original white-box attack. Once the model changes, the attack no longer satisfies the white-box criteria, undermining the validity of such an evaluation.
>
> > Regarding the experimental results, the authors did not provide more insightful hypotheses or validations compared to the original paper. For instance, there is no deeper exploration of why FViT performs poorly on ImageNet-C or why the Diffusion model used in the original work outperforms Stable Diffusion.
>
> We point out that the original diffusion model is the only one trained on the original dataset, which is the most likely explanation for the performance discrepancy. We make an insightful and logical hypothesis that DDS will struggle with samples that are adversarial for both the diffusion model and the classification model.
>
> > On page 6, section 3.4 (Extensions), sub-section 1, second line, there is a redundant phrase "computational cost," which appears to be a typographical error.
>
> We fixed the typographical error.
>
> > Could you please further explain why the proposed new aggregation method is better than the original one. For example, what is the meaning of saturated probability distribution and is there any evidence to show that the original aggregation does struggle with saturated probability distribution. Moreover, why the new aggregation method has more descriptive top-k accuracy?
>
> We performed experiments that compared our aggregation method and the original aggregation method and added the results in the appendix.

---

### Decision · Action_Editor_V66q · 2025-06-17

**Recommendation:** Accept with minor revision

**Additional Comments:**

The paper demonstrates good technical quality in its core reproduction effort, described as thorough and rigorous by reviewers, and is generally well-written and clear. However, the reproduction effort is noted to be non-trivial but not particularly challenging given the original code availability. The extensions, while broadening the scope, are criticized for lacking depth in analysis and providing only incremental or trivial insights without sufficient explanation or surprising findings. Issues like figure attribution and appendix presentation also need to be fixed before publishing.

**Audience:**

Yes

**Audience Explanation:**

Yes, some individuals in TMLR's audience would likely be interested. Researchers working on interpretability and robustness of Vision Transformers, particularly those building upon or using the DDS/FViT method, would find value in the independent verification of the original paper's claims, the demonstration of its applicability across ViT, DeiT, and Swin architectures, and the empirical comparison involving Stable Diffusion. The observation regarding DDS's poor performance on non-adversarial corruptions (ImageNet-C) and its potential overfitting to adversarial attack styles, identified as novel and important by one reviewer, would also be of specific interest to this community.

**Claims And Evidence:**

Yes

**Claims Explanation:**

The evidence supporting the reproduction of the original paper's core findings is generally assessed as accurate, convincing, and thorough by multiple reviewers, who commend the meticulous reproduction and technical rigor. However, the evidence for the extended claims (e.g., applicability to DeiT/Swin, performance of alternative diffusion models like Stable Diffusion, poor performance on ImageNet-C) is less consistently convincing; while the results themselves are presented, reviewers find the interpretation and analysis supporting why these results occur (e.g., why DDS fails on ImageNet-C, why Stable Diffusion underperforms) to be superficial, lacking deeper hypotheses or validation, and the insights derived are deemed relatively trivial or expected.